# Enhancing Methane Recovery with Cryogenic Liquid CO_2_ Cyclic Injection: Determination of Cyclic Injection Parameters

**DOI:** 10.3390/ijerph192013155

**Published:** 2022-10-13

**Authors:** Duo Zhang, Shixing Fan

**Affiliations:** 1College of Safety Science and Engineering, Xi’an University of Science and Technology, Xi’an 710054, China; 2Department of Energy Engineering, Xi’an University of Science and Technology, Xi’an 710054, China

**Keywords:** CO_2_-ECBM, freeze–thaw, liquid CO_2_, permeability enhancements, engineering application, gas extraction

## Abstract

Carbon dioxide (CO_2_) is both a primary greenhouse gas and a readily available energy source. In this study, a new underground coal permeability enhancement technique utilizing cryogenic liquid CO_2_ (L-CO_2_) cyclic injection is proposed. The key parameters that determine the feasibility of this technique are cycle period and cycle number within a fixed working period. The optimal value of these two parameters mainly depends on the pore structure evolution law of coal cores before and after cryogenic L-CO_2_ cyclic freeze–thaw. Accordingly, nuclear magnetic resonance (NMR) was employed to study the evolution characteristics of the fracture networks and pore structures in coal cores subjected to different freeze–thaw cyclic modes. The results demonstrated that the amplitude and width of all peaks of the *T*_2_ spectra of the three coal cores (lignite, gas coal, and 1/3 coking coal) increased with an increase in the number of injection cycles. Additionally, as the number of freeze–thaw cycles (*N*_c_) increased, the total porosity and effective porosity of the coal cores increased linearly before stabilizing, while the residual porosity first steadily diminished and afterwards became constant. Furthermore, the variation in the total porosity and residual porosity of the coal cores continuously diminished with an increase in the level of metamorphism. The NMR permeability of the coal cores showed a similar pattern to the porosity. Accordingly, the optimal parameters for cryogenic L-CO_2_ cyclic injection during a complete working time were determined to be *N*_c_ = 4 and *P*_c_ = 30 min. A field test demonstrated that after L-CO_2_ cyclic freeze–thaw treatment, the average gas drainage concentration of a single borehole in the test region increased by 1.93 times, while the pure flow of a single gas drainage borehole increased by 2.21 times. Finally, the gas attenuation coefficient decreased from 0.036 to 0.012. We concluded that the proposed permeability enhancement technique transformed coal seams from hard-to-drain to drainable.

## 1. Introduction

Gas disasters are one of the most common factors affecting coal mine safety [1,2]. Roughly half of the coal seams in China have a high gas content, and over 70% of coal mines have generally high contents of coalbed methane (CBM) [3,4]. This far-reaching distribution of high-gas and low-permeability coal seams represents an immense danger to mine security. Furthermore, methane, the fundamental part of coalbed methane, is a major source of severe environmental pollution [5,6]. The annihilation effect of methane on the ozone layer and its greenhouse impact are seven times and twenty-one times higher than those of carbon dioxide (CO_2_), respectively [7,8]. However, CBM can be utilized as a spotless, productive, and environment-friendly wellspring of energy. Therefore, enhancing CBM drainage efficiency can not only prevent gas hazards in coal mines but can also help in full utilization of underground resources and protection of the atmosphere. However, the low permeability and dense matrix characteristics of most high-gas coal seams in China significantly hinder the efficiency of CBM recovery [2,4,9].

Over the last few decades, numerous hydraulic measures and stimulation treatments have been carried out in coal mines to improve the permeability of coal seams and enhance gas desorption. Among these, hydraulic fracturing (HF) is most commonly used for enhanced coal bed methane (ECBM) extraction [10]. However, the increase in coal surface moisture caused by HF severely blocks the desorption of methane. Moreover, conventional water-based hydraulic fracturing consumes a huge amount of water and can even pollute underground water resources. Thus, scholars have suggested several non-aqueous fracturing technologies, primarily techniques using CO_2_ in various phase states (liquid, gas, and supercritical state) as fracturing fluids [11,12].

Compared with HF, liquid CO_2_ fracturing (L-CO_2_) has the advantageous characteristics of a low temperature (−35 °C) [13], phase-transition pressurization (liquid–gas phase transition volume ratio of 1:557 at 1 atm) [14], high latent heat of vaporization (347 kJ/kg) [15], low viscosity [16], powerful adsorption capacity [17], and no water sensitivity [18], which have been widely applied in the studies on ECBM recovery. Other than these factors, injecting liquid CO_2_ into non minable coal seams to displace CH_4_ was also helpful to CO_2_ storage, and lay a foundation for the realization of the long-term goal of “carbon peaking and carbon neutralization”. Recently, many CO_2_-ECBM pilot projects have been successfully conducted in North America and Australia [19]. The results of these projects demonstrated that L-CO_2_ can effectively increase the production of CH_4_ and that CO_2_ can be geologically sequestered. In China, although L-CO_2_ fracturing has only recently been introduced, some micro-pilot tests of CO_2_-ECBM have been carried out in the Ordos and Qinshui Basin [20].

Previous studies have mainly focused on the effects of the huge phase energy of L-CO_2_ on the coal body. Chen et al. [21] derived the sphere of influence of this technology by combining numerical simulation and underground industrial experiments. Cao et al. [22] conducted an industrial test of the L-CO_2_ phase-change fracturing technology in Luan coalfield and found that the permeability and methane drainage effectiveness in the L-CO_2_ fracturing boreholes were significantly improved. Lu et al. [23] determined that using L-CO_2_ as a fracturing fluid reduced the fracture initiation pressure (FIP) by more than 30%. Jiang et al. [24] developed an L-CO_2_ phase-change gas jet pressure model, which was used to explore the mechanism of L-CO_2_ phase jet coal breaking and fracture expansion. Yin et al. [15,25] concentrated on the effects of geological and engineering factors on induced fracture propagation by performing hydraulic and L-CO_2_ fracturing experiments on coal specimens gathered from the South Sichuan Basin in China.

In this study, in contrast to the L-CO_2_ phase-change fracturing technology, a novel ECBM extraction technology, based on cryogenic L-CO_2_ cyclic injection, is proposed, as shown in Figure 1 [26], which comprises a liquid storage tank, remote control system, and remote monitoring system. The kernel of this technology comprises the frequency conversion cryogenic L-CO_2_ piston pump. Upon first injecting gaseous CO_2_ into the borehole, the drilling pressure increases to 2.0 MPa. L-CO_2_ is then injected into the borehole using an L-CO_2_ piston pump until the drilling pressure increases to 5.0–6.0 MPa. The entire injection duration is approximately 20 min. Then, the L-CO_2_ pipeline valve and piston pump are turned off, and the drilling pressure begins to decline. When the drilling pressure drops to 2.0 MPa, the L-CO_2_ pipeline and pump are opened again. Next, the above-mentioned steps are all repeated, and the interval between two consecutive pumping injections is termed the “period of cycle”. Related field applications of this technology have been performed in the Huainan and Weibei coalfields, which demonstrated its effectiveness in terms of higher methane production [4,8,9]. However, there is a certain blindness in determining the key parameters of L-CO_2_ cyclic injection, which extremely limits the promotion and application of this technology in the permeability enhancement of low permeability coal seams.

During the injection process of L-CO_2_, the coal body encompassing the injection boreholes undergoes cooling (by cryogenic L-CO_2_) and heating (by geothermal) repeatedly, during which the fracture networks and pore structures of coal are altered [27,28]. However, the evolution characteristics of fracture networks and pore structures of coal under the effects of cyclic freeze–thaw are inadequately perceived. Therefore, in this study, a cryogenic L-CO_2_ cyclic freeze–thaw experimental system was developed, and three types of coal samples were exposed to cryogenic L-CO_2_ cyclic freeze–thaw treatment. Meanwhile, nuclear magnetic resonance (NMR) was employed to investigate the variations in the characteristics of fracture networks and pore structures under the effect of cryogenic L-CO_2_ cyclic injection. Finally, a pilot test using the technology was conducted in the Zhangji coal mine, based on the cyclic freeze–thaw experimental results.

## 2. Experimental Section

### 2.1. Specimen Preparation

Three differently ranked coal blocks were obtained from the Da Nanhu Coal Mine (DNH Lignite), Zhaolou Coal Mine (ZL Gas coal), and Zhangji Coal Mine (ZJ 1/3 Coking coal). The geographic locations of the mines are shown in Figure 2. Table 1 lists the sizes of the above-mentioned three coal samples and the results of proximate and ultimate analyses.

Cores with a diameter of 10 mm and length of 20 mm were then drilled from the coal blocks using a Coring machine. The end-faces of the cores were polished using a smoothing machine. Subsequently, the densities and sonic wave velocities were determined, and coal cores with similar properties were selected for freeze–thaw experiments.

### 2.2. Experimental System and Equipment

Figure 3 shows the schematic of the cryogenic L-CO_2_ freeze–thaw cyclic experimental system, which mainly comprises five components: an L-CO_2_ Dewar bottle, a low-temperature piston pump, a freeze–thaw container, data monitoring equipment, and an NaOH solution device. The piston pump was used to inject the cryogenic L-CO_2_ into the container. The container was wrapped with a thermal insulation layer to prevent vaporization of the L-CO_2_. Meanwhile, an electrical pressure gauge and temperature sensor with a suitable range were chosen for installation in the container for monitoring the pressure and temperature. Based on this, the state of CO_2_ could be determined according to the phase transition diagram of CO_2_. As a safety insurance, a safety valve was installed on the outlet of the freeze–thaw container to release CO_2_ gas when the pressure surpassed 5.2 MPa.

During the experiment, nuclear magnetic resonance (NMR, MicroMR23-025V Niumag Co., Suzhou, China) spectroscopy was employed to analyze the fracture networks and pore structures before and after the treatment of coal samples with cryogenic L-CO_2_. The MicroMR23-025V generated a main magnetic field of 0.51 T. By fitting the attenuated signals of spin-echo strings, acquired from a test of CPMG pulse sequences on coal cores entirely saturated with water, a distribution curve of transverse relaxation time *T*_2_ could be obtained [29]. The relationship between *T*_2_ and the ratio of pore surface to pore volume can be expressed as Equation (1) as follows:(1)$$\frac{1}{T_{2}}=\rho {\left(\frac{S}{V}\right)}_{pore}=F_{s}\cdot \rho \cdot \left(\frac{1}{r}\right)$$
where *ρ* is the surface relativity of mineral surface (μm/s); (*S*/*V*)*_pore_* indicates the surface area to volume (μm^−1^); *F*_s_ denotes the shape factor of pores, for spherical pore, *F_s_* = 3; *r* is the pore radius (μm).

As described by Qin et al. [30,31], the characteristics of pores can be inferred on the basis of amplitudes, peak areas, and continuity of the *T*_2_ distribution curve. Moreover, the *T*_2_ value corresponds to the distribution of different pore sizes. The micropores, mesopores, and macropores can be divided using the *T*_2_ values of 10.0 and 100 ms [32].

A vacuum drying oven (DZF-6050, Yiheng Apparatus Co., Shanghai, China), vacuum water saturation device (ZYB-II, Huaxing Petroleum Devices Co., Nantong, China), rock centrifuge (Sorvall Legend, Thermo Fisher Scientific Co., Waltham, MA, USA), and ion sputter coater (GVC-2000, Microhezao Co., Shanghai, China) were also used as auxiliary equipment, along with the NMR instruments during the test period.

### 2.3. Experimental Procedures

Previous studies have shown that cryogenic L-CO_2_ cyclic freeze–thaw can promote the propagation of fracture networks and development of pore structures of coal seams. In a field test, the injection of L-CO_2_ usually needs to be completed within a certain fixed working period. In general, this fixed working period lasts approximately 4 h in most mining areas of China. Accordingly, the total time for each cyclic freeze–thaw experiment was determined to be 240 min. In addition, prior field tests in the Huainan and Hancheng mining areas had shown that the injection pressure of liquid CO_2_ cannot exceed 6.0 MPa, owing to the limitations of the size of the borehole and current equipment capabilities. Therefore, in order to ensure a consistent pressure in the field injection pressure, the injection pressure was set to 5.0 MPa in the experiment.

Accordingly, the detailed experimental procedures were as follows:

(a) Before the formal test, all three different rank coal cores were weighed and then placed in a vacuum drying oven at 60 °C for 8 h. This was regarded as the initial condition. Subsequently, NMR pre-testing was performed on the cores under water-saturated and irreducible water conditions.

(b) Then, the cores were wrapped with 200 mesh fine gauze and placed in a self-designed freeze–thaw container. Subsequently, L-CO_2_ was injected into the container using four different cyclic injection modes, as shown in Figure 4. Each injection mode comprised a different number of freeze–thaw cycles (*N*_c_). A complete freeze–thaw cycle included a period of freezing (*t_f_*) and the same period of thawing at room temperature (*t_r_*). In this study, *t_f_* was always equal to *t_r_* in each injection mode, i.e., *t_f_* = *t_r_*. The sum of *t_f_* and *t_r_* was called the period of cycle (*P*_c_). Mode I, shown in Figure 4, included a total of two freeze–thaw cycles and each cycle lasted 120 min. Similarly, the number of cycles (*N*_c_) for modes II, III, and IV was equal to 3, 4, and 8, respectively, while the *P*_c_ of modes II, III, and IV lasted 80, 60, and 30 min, respectively. In conclusion, in each method, *N*_c_ multiplied by the *P*_c_ was equal to 240 min.

In each experiment, valves 1, 3, and 4 were opened in sequence until the temperature in the container dropped to −8.0 to −10.0 °C; then, valve 4 was gradually closed until the pressure in the container rose to approximately 2.0 MPa. Subsequently, valve 1 was closed, valve 2 was opened, and the booster pump started to inject L-CO_2_ into the container until the pressure of the container rose to approximately 5.0 MPa; then, valve 3 was closed. In addition, the set pressure of the pressure relief valve was 5.2 MPa. When the pressure in the vessel exceeded 5.2 MPa, the safety valve was activated to maintain a pressure of 5.0–5.2 MPa in the vessel.

(c) After the freeze–thaw treatments of L-CO_2_, NMR testing was performed on all the cores under water-saturated and irreducible water conditions again. Simultaneously, their weights were recorded under both conditions using an electric balance. Then, the results of the NMR test were used to analyze the evolution of the crack networks and pore structures.

## 3. Evaluation of the Experimental Data

### 3.1. Porosity Calculated from the T_2_ Spectra

As indicated by the exploratory methods in Section 2.3, three differently ranked coal cores were subjected to NMR tests under water-saturated and irreducible water conditions after each cryogenic L-CO_2_ freeze–thaw treatment to obtain the T_2_ spectra of the cores in the two states. In this study, in order to quantify the variation of porosity in cores before and after the cryogenic L-CO_2_ cyclic freeze–thaw treatment, the porosity of the cores was characterized using the approximation of the NMR porosity (*φ_t_*) [33]. The values were determined by the weighted porosity, which could be calculated using the 100% water saturation method [34].

According to the water saturation method, the total porosity (*φ_t_*) is equal to the signal magnitude of liquid ^1^H in water-saturated coal, which could be approximately calculated by estimating the water volume in coal. Theoretically, the *φ_t_* represents the pore volume fractions of all the water-bearing pores [35], and the water-bearing pores are mainly composed of the pores occupied by bound water and the pores occupied by free water. Therefore, the residual porosity (*φ_r_*) can be defined as the bound water fraction, and the effective porosity (*φ_e_*) corresponds to the free water fraction [36]. The relationship between porosities can be expressed by Equations (2) and (3):(2)$$\varphi_{r}=\varphi_{t}\times \frac{BVI}{BVI+FFI}$$
(3)$$\varphi_{e}=\varphi_{t}\times \frac{FFI}{BVI+FFI}$$
where *BVI* and *FFI* is the bound fluid and free fluid indexes, respectively; *BVI* + *FFI* represents the total fluids index; *BVI* can be calculated by the total *T*_2_ spectrum area fraction under centrifuge condition, while *BVI* + *FFI* can be calculated by the total *T*_2_ spectrum area fraction under water saturation conditions [37], and the *FFI* is obtained by subtracting the BVI from the *BVI* + *FFI*.

### 3.2. NMR Permeability

The permeability of a coal seam is an important index reflecting the ability of gas to flow through, which has a close relationship with the distribution and connectivity of pores. According to previous studies [9], the permeability of the coal seam can be calculated by the porosity parameters and the proportion of the pore sizes derived from the NMR experiment. Due to its higher accuracy, the SDR model was chosen to assess the permeability of the coal seam in this study. This equation developed by Kenyon et al. [38] can be expressed as:(4)$$k_{S}=\alpha \cdot \varphi^{m}\cdot {\left(T_{2gm}^{a}\right)}^{n}$$
where *α*, *m*, and *n* are the empirical constants related to the feature of coal rock masses, and *φ* is the porosity of coal. $T_{2gm}^{a}$ is the *T*_2_ geometric mean of coal samples under water-saturated conditions.

After the regression calculation, the above-mentioned equation can be expressed as Equation (5) [39]:(5)$$k_{S}=0.0224\cdot {\left(T_{2gm}^{a}\right)}^{1.534}\cdot {\left(T_{2gm}^{b}\right)}^{0.182}$$
where $T_{2gm}^{a}$ and $T_{2gm}^{b}$ are the *T*_2_ geometric means for coal samples under water-saturated and irreducible water conditions, respectively, which can be determined using Equation (6) [40]:(6)$$T_{2gm}=\exp(\sum_{T_{2i}}^{T_{2\max}}\frac{A_{j}}{A_{c}}ln(T_{2j}))$$
where *T*_2*i*_ and *T*_2max_ are the initial and maximum values, respectively, *T*_2*i*_ = 0.01 ms, *T*_2max_ = 10,000 ms. *A_j_* is the amplitude corresponding to *T*_2*j*_, *A_c_* represents the cumulative amplitude of the whole *T*_2_ spectrum, and *T*_2*i*_ is the individual *T*_2_ value.

## 4. Results and Discussion

### 4.1. Variation in the Temperature and Pressure of the Freeze–Thaw Container

Figure 5 shows the temperature and pressure changes in the freeze–thaw container with time under four different injection modes. Figure 5 shows that the liquid CO_2_ cyclic injection was carried out using four different modes with the starting point of the pressure curve in each circulating container at approximately 2.1–2.3 MPa. The pressure change in the container during the subsequent injection process was roughly the same as predicted in Figure 4. This was because the gas pressure in the Dewar was approximately 2.2 MPa, and each injection of liquid CO_2_ was carried out in the order of “gas injection–gas–liquid balance–liquid injection”. The temperature in the container dropped to −23–−24 °C within 3 min after the start of each cycle of liquid injection and was maintained at this temperature during the high-pressure stage. When the pressure was released, the temperature gradually increased. In each injection mode, the value of the temperature rise of the next cycle was lower than that of the previous cycle.

### 4.2. T_2_ Spectrum Analysis of Coal Samples before and after L-CO_2_ Treatment

As indicated in Section 2.2, the *T*_2_ spectrum of the coal cores could reflect the pores change qualitatively. The T_2_ values ranging from 0.1 to 10 ms correspond to the micropores, also known as adsorbed pores, providing space for the adsorbed gas [33]. The *T*_2_ values larger than 10 ms indicate the presence of mesopores, macropores, or microfractures, referred to as “seepage pores”, which provide migration channels for the free gas [34].

Figure 6 shows the *T*_2_ spectra of three types of coal cores (lignite, gas coal, and 1/3 coking coal) submerged in water-saturated and irreducible water conditions, before and after the coal cores were treated with various cryogenic L-CO_2_ cyclic freeze–thaw modes. Figure 6a,c,e shows that under the water-saturated condition, the *T*_2_ spectra of lignite and gas coal cores treated with various freeze–thaw methods all contained three peaks, while the *T*_2_ spectra of 1/3 coking coal cores contained two peaks. Upon comparing the *T*_2_ spectra of each coal core treated with various cryogenic L-CO_2_ treatment modes, the results showed that the amplitude and width of all peaks of the *T*_2_ spectra for the three coal samples increased with an increment in the number of cycles (*N*_c_) under water-saturated conditions. However, Figure 6b,d,f shows that the *T*_2_ spectra of three kinds of coal cores all contained two peaks under the irreducible water condition after cryogenic L-CO_2_ treatment. Moreover, as the *N*_c_ increased, the amplitude and width of the *T*_2_ spectra for each coal sample gradually diminished. In summary, when the *N*_c_ surpassed 4, the peak change amplitude of the *T*_2_ spectrum decreased under the two states.

### 4.3. Porosity Change in Coal Samples before and after L-CO_2_ Treatment

As the density of water (*ρ_w_*) is close to 1 g/cm^3^, the *φ_t_* of the coal core under saturated water conditions can be calculated using Equation (7):(7)$$\varphi_{t}=\frac{V_{w}}{V_{\mathrm{core}}}=\frac{m_{s}-m_{d}}{\rho_{w}\cdot V_{core}}\approx \frac{m_{s}-m_{d}}{V_{core}}$$
where *V_w_* is the volume of water in the coal core under saturated conditions, *V_core_* is the volume of coal cores under saturated conditions, *m*_s_ is the mass of coal cores after 48 h of saturated conditions, *m_d_* is the mass of coal cores under dry conditions, and *V_core_* is the volume of the coal core.

Based on Equation (7) and the theory outlined in Section 3.1, the normalized value of the cumulative amplitude of the *T*_2_ spectra in the water-saturated condition can be regarded as *φ_t_* of the coal core [41]. Correspondingly, the maximum cumulative amplitude of the *T*_2_ spectrum under the irreducible water condition is the *φ_r_* of the coal core. The *φ_e_* is the difference between the *φ_t_* and the *φ_r_*. Figure 7 shows the process for the calculation of porosity percentage and cumulative porosity of 1/3 coking coal in the original state and after treatment with mode III (*N*_c_ = 4, *P*_c_ = 60 min) under water-saturated and irreducible water conditions. As shown in Figure 7, the *φ_t_* and *φ_e_* of the coal sample can be calculated using Equations (2), (3) and (7). Using the same method, the *φ_t_* and *φ_e_* of three kinds of coal samples treated with four different freeze–thaw cyclic modes could be obtained. See Appendix A for the calculation results.

Figure 8 depicts the variation in the *φ_t_* of the three types of coal cores after four different freeze–thaw cyclic modes. Figure 8 shows that with an increase in the *N*_c_, the *φ_t_* of the three types of coal samples first increased linearly before gradually stabilizing. That is, when the *N*_c_ > 4, the *φ_t_* of the three types of coal samples remained basically unchanged. In the case of *N*_c_ < 4, the linear fitting method was used to analyze the relationship between the *φ_t_* of the three types of coal samples and the *N*_c_. The related parameters of the fitting functions between *φ_t_* and *N*_c_ are shown in Table 2.

It can be seen from Table 2 that as the degree of coal sample metamorphism increased, the slope of the fitted line (the value of a) decreased gradually. That is, the increasing rate of *φ_t_* with the *N*_c_ gradually decreased with an increase in coal metamorphism. This was because freeze–thaw cycles promoted the development of various scale pores inside the coal cores, especially micropores. However, with an increase in the *N*_c_, the duration of “freezing” and “thawing” experienced by the coal core in each cycle decreased. As shown in Figure 9, in each freeze–thaw cyclic mode, the temperature of the coal core rose after the *i* + 1 cycle was less than the temperature rise after the i cycle (Δ*T_i_*_+1_ < Δ*T_i_*). This was because the Δ*T_i_* varied with the *N*_c_ and showed a gradual decrease. As a result, the promotion effect of freeze–thaw cycles on the development of coal core porosity gradually weakened. Thus, when the *N*_c_ exceeded 4, the *φ_t_* growth effect of the coal sample became less obvious.

As indicated by Equations (2) and (3), the variation in rules of *φ_e_* and *φ_r_* for the three types of coal cores after treatment with four different cycles of freeze–thaw are shown in Figure 10. It can be seen from Figure 10 that the variation in the *φ_e_* of three types of coal cores was positively correlated with increase in *N*_c_, while that in the *φ_r_* was negatively correlated with increase in *N*_c_. Similar to the variation in *φ_t_*, when *N*_c_ < 4, as the *N*_c_ increased in increments, *φ_e_* increased and *φ_r_* diminished linearly. When *N*_c_ > 10, both the *φ_e_* and *φ_r_* remained fundamentally unaltered.

### 4.4. NMR Permeability Change Analysis

In view of the *T*_2_ spectra of the three types of coal cores shown in Figure 6, the NMR permeability of the three different rank coal cores treated with four different cyclic freeze–thaw modes were determined using Equation (5). The variation in NMR permeability with an increment in the *N*_c_ is displayed in Figure 11. Overall, the varying trend of NMR permeability of the three different types of coal cores was consistent with the varying trend of the *φ*_t_ and *φ*_e_ of the coal cores; that is, when *N*_c_ < 4, the NMR permeability of coal cores increased linearly with the *N*_c_. Contrastingly, when *N*_c_ ≥ 4, the NMR permeability of coal cores remained stable. Additionally, the order of the increase in NMR permeability for the three coal cores was as follows: lignite > gas coal > 1/3 coking coal, which was consistent with the order of the growth rates of the total porosity and *φ_e_* of the three coal samples.

In summary, when the *N*_c_ exceeded 4, the *φ_e_* and NMR permeability of the three types of coal cores no longer increased. Based on the above-mentioned analyses, in order to achieve a more remarkable permeability enhancement effect during a complete working time, the optimal parameters for cryogenic L-CO_2_ cyclic injection during a complete working time were determined to be *P*_c_ = 30 min and *N*_c_ = 4.

## 5. Field 

### 5.1. Field Situation

To confirm the permeability improvement impact of the cryogenic L-CO_2_ cyclic injection, a field test was conducted in the No. 6 coal seam of the Zhangji coal mine, located in Huainan City, Anhui Province. The No. 6 coal seam was identified as an outburst coal seam, with an average thickness of 4.5 m and a dip angle of 28°–31°. The average gas content and gas pressure were 7.08 m^3^/t and 1.25–1.69 MPa, respectively. In addition, the Protodyakonov coefficient *f* of the No. 6 coal seam was 0.204. The coal seam has poor permeability and poor gas drainage.

### 5.2. Borehole Layout

The test boreholes were situated in the intake airway of the working face, 17,246. Gas drainage tests of boreholes after cryogenic L-CO_2_ cyclic injection in the coal seam were conducted. The dig angle and the azimuth angle of boreholes were 30° and 90°, respectively. The design diameter of the boreholes was 93 mm, and the length was 140 m. The spacing between adjacent boreholes was 5.0 m, and the sealing length of each borehole was designed to be 40 m. In order to ensure that the stress state of the coal seam in the drilling area was not affected by mining of the working face, the boundary of boreholes was kept more than 200 m away from the working face.

The boreholes were divided into three groups by two L-CO_2_ injection boreholes (7# and 14#), where boreholes from 1# to 6#, boreholes from 8# to 13#, and boreholes from 15# to 20# were marked as Group 1, Group 2, and Group 3, respectively. Furthermore, another 18 boreholes near this area were chosen as the reference group. The borehole layout is shown in Figure 12.

### 5.3. Test Results and Discussion

In the field test, L-CO_2_ was injected into the 7# and 14# boreholes according to mode III. The entire injection process lasted 240 min. The variation in the pressure of injection boreholes is shown in Figure 13. When the drilling pressure increased to 5.0 MPa for approximately 20 min, the cryogenic L-CO_2_ piston pump was turned off until the drilling pressure decreased to 2.5 MPa. Subsequently, the cryogenic L-CO_2_ piston pump was turned on again to keep the pressure increasing to 5.0 MPa. Next, the boreholes were injected again following the order of “injection-holding pressure-injection-holding pressure” until the desired number of cycles was achieved. Finally, the drainage boreholes were connected to the main gas-extraction pipe.

The gas extraction concentration and flow of the drainage boreholes were observed for 36 days, as shown in Figure 14a. The gas-drainage concentration in the reference area was 25.5%, whereas that of the drainage boreholes in the injection area was 49.2% (average), which was 1.93 times higher. Similar to the gas concentration, the gas extraction amount of drainage boreholes in the injection area increased from 0.022 m^3^/min to 0.048 m^3^/min, which was 2.21 times the amount of drainage boreholes in the reference area.

In addition, according to the China National Code for Coal Mine Gas Drainage AQ 1027-2006 [42,43], the gas drainage attenuation coefficient could be used as an index to evaluate the degree of difficulty of pre-drainage in the coal seam, which could be calculated using Equation (8):(8)$$Q_{t}=Q_{0}\cdot e^{-\beta \cdot t}$$
where *Q*_0_ is the initial flow of the drainage hole, *t* is the drainage time, *Q_t_* is the gas drainage flow at time *t*, and *β* is the attenuation coefficient of the borehole gas flow. According to Equation (8), the attenuation coefficient could be obtained by fitting the average gas drainage flow of a single borehole. As shown in Figure 14b, the attenuation coefficient of the gas amount decreased from 0.036 to 0.012 after the cryogenic L-CO_2_ injection. This indicated that the original coal seam was previously hard to drain but had become drainable. As a result, the gas drainage efficiency improved significantly, which will be helpful in reducing the gas content, improving the permeability of the coal seam, and eliminating coal and gas outbursts.

## 6. Conclusions

(1) This study proposed a novel ECBM extraction technology, based on the cryogenic L-CO_2_ cyclic injection. The kernel of this technology is to scientifically determine the *N*_c_ and *P*_c_ of cryogenic L-CO_2_ cyclic injection within a fixed working time.

(2) According to NMR, in the water-saturated condition, and the amplitude and width of all peaks of the *T*_2_ spectra of the three coal samples increased with an increase in the *N*_c_. In the irreducible water condition, the *T*_2_ spectra of all three coal samples contained two peaks, and as the *N*_c_ increased, the amplitude and width of the *T*_2_ spectra gradually decreased. Under both conditions, when *N*_c_ ≥ 4, the peak amplitude of the *T*_2_ spectrum decreased.

(3) As the *N*_c_ increased, the *φ_t_* and *φ_e_* of the three coal samples first increased linearly and then gradually stabilized, while, the *φ_r_* gradually decreased at first and then stabilized. In addition, as the degree of coal sample metamorphism increased, the *φ_t_* and *φ_r_* variation rate of the coal sample gradually decreased.

(4) The field test in the Zhangji coal mine showed that the average single hole gas drainage concentration increased from 25.5% to 49.2% after cyclic cryogenic L-CO_2_ injection, representing a 1.93-fold increase. The average single hole gas drainage pure flow increased from 0.022 m^3^/min to 0.048 m^3^/min, showing a 2.21-fold increase. The attenuation coefficient of gas drainage decreased from 0.036 to 0.012, and the coal seam transformed from hard-to-drain to drainable.

## Figures and Tables

**Figure 1 ijerph-19-13155-f001:**
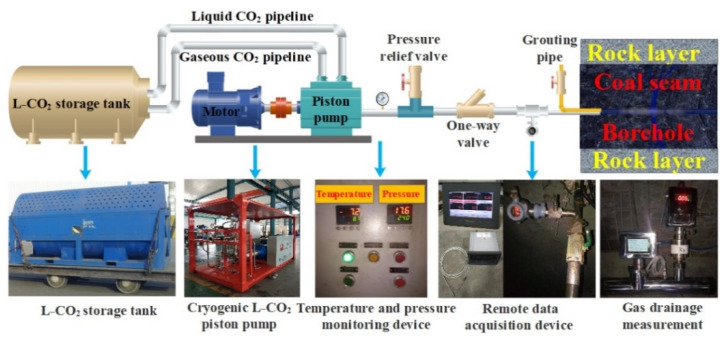
Schematic of the cryogenic L-CO_2_ cyclic injection system and equipment.

**Figure 2 ijerph-19-13155-f002:**
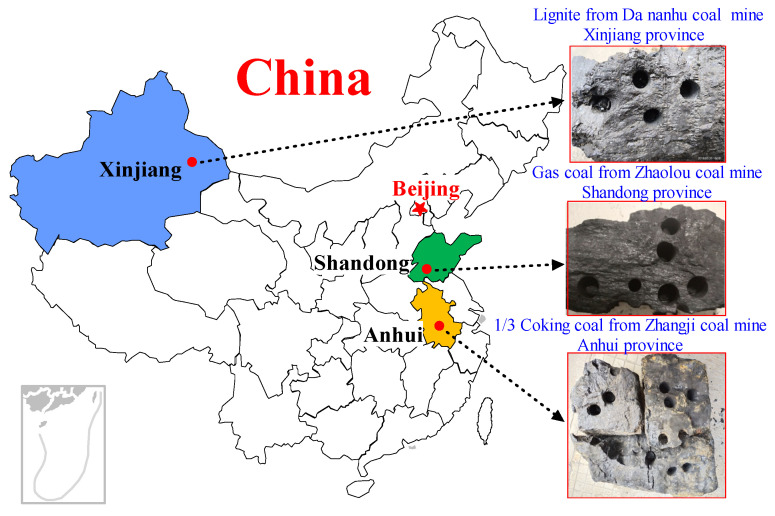
Geographic location and preparation of the three coal cores in this study.

**Figure 3 ijerph-19-13155-f003:**
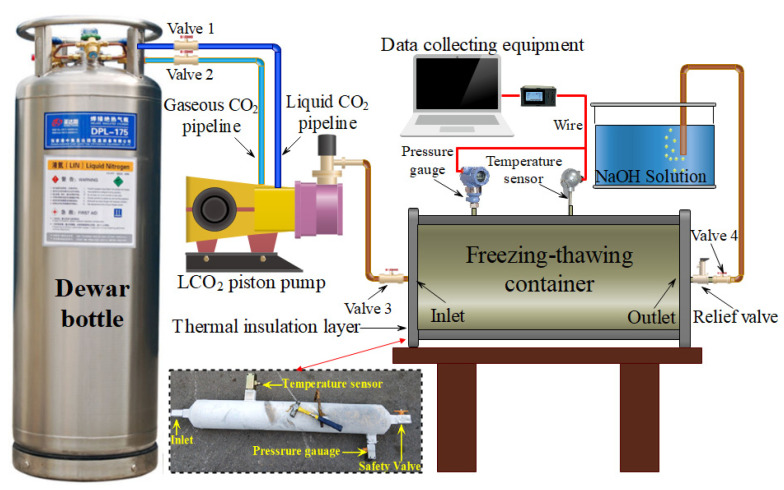
Diagram of the cryogenic L-CO_2_ cyclic freeze–thaw experimental system.

**Figure 4 ijerph-19-13155-f004:**
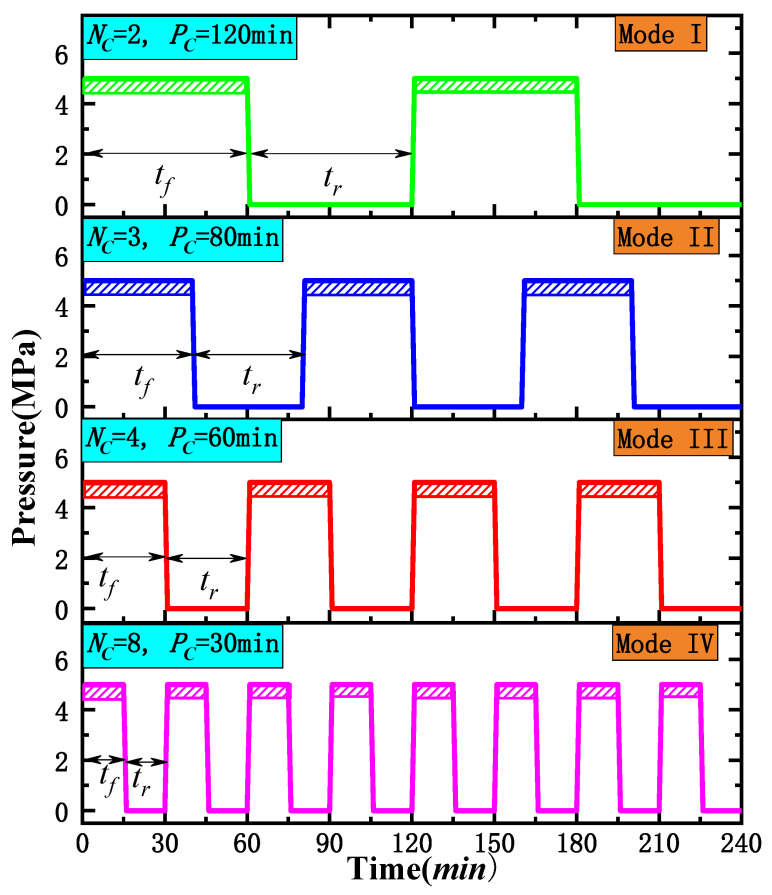
Modes of cryogenic L-CO_2_ cyclic injection. (Note: *N*_c_ represents Number of Cycle, and *P*_c_ represents Period of Cycle).

**Figure 5 ijerph-19-13155-f005:**
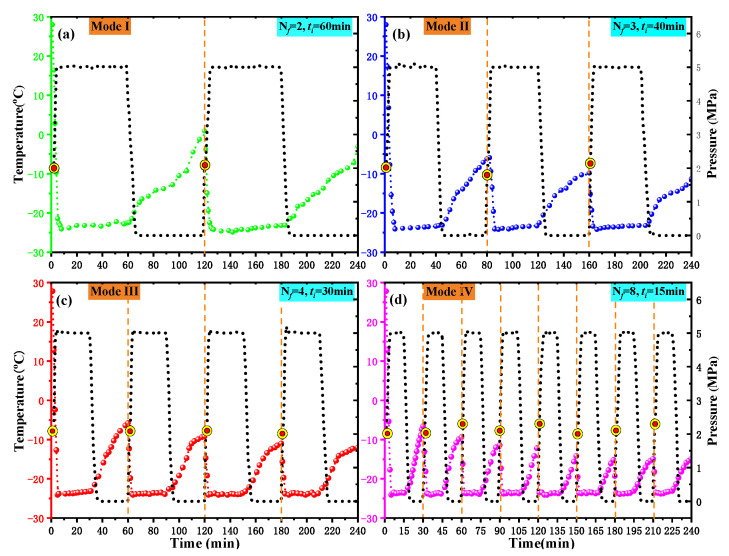
Variation in the temperature and pressure of the freeze–thaw container. (**a**) Model I. (**b**) Model II. (**c**) Model III. (**d**) Model IV. (Note: *N*_c_ represents Number of Cycle, and *P*_c_ represents Period of Cycle).

**Figure 6 ijerph-19-13155-f006:**
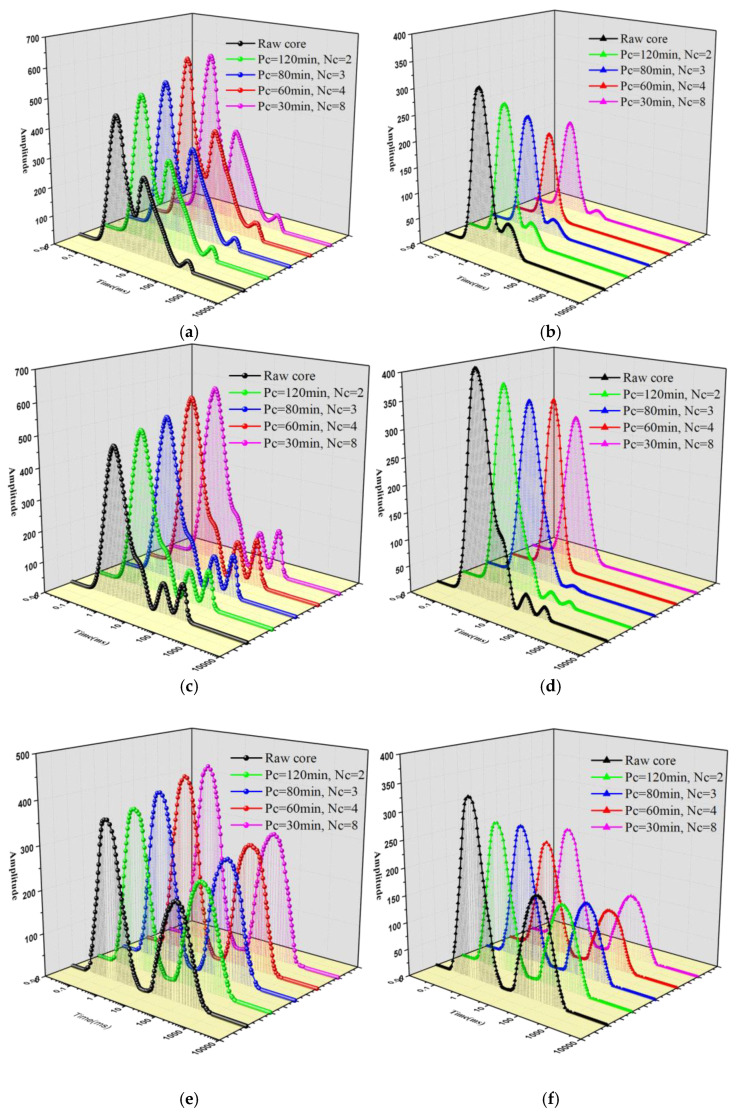
*T*_2_ spectra of lignite, gas coal, and 1/3 coking under water-saturated and irreducible water conditions before and after coal cores were treated with various cryogenic L-CO_2_ cyclic freeze–thaw methods. (**a**) *T*_2_ spectra of lignite under water-saturated condition. (**b**) *T*_2_ spectra of lignite under irreducible water condition. (**c**) *T*_2_ spectra of gas coal under water-saturated condition. (**d**) *T*_2_ spectra of gas coal under irreducible water condition. (**e**) *T*_2_ spectra of 1/3 coking coal under water-saturated condition. (**f**) T_2_ spectra of 1/3 coking coal under irreducible water condition.

**Figure 7 ijerph-19-13155-f007:**
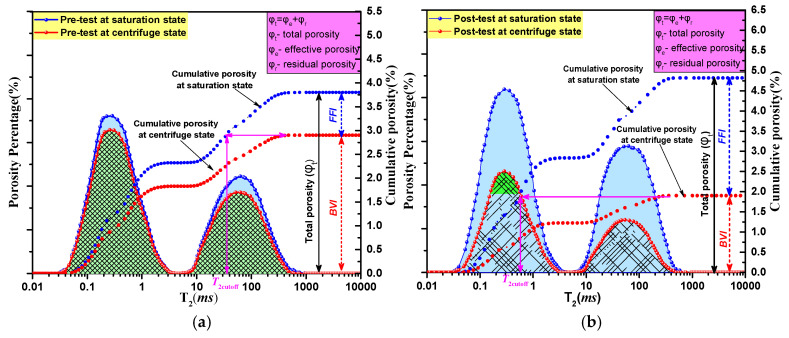
Schematic of the porosity percentage and cumulative porosity of 1/3 coking coal under water-saturated and irreducible water conditions: (**a**) Original state (**b**) *P*_c_ = 60 min.

**Figure 8 ijerph-19-13155-f008:**
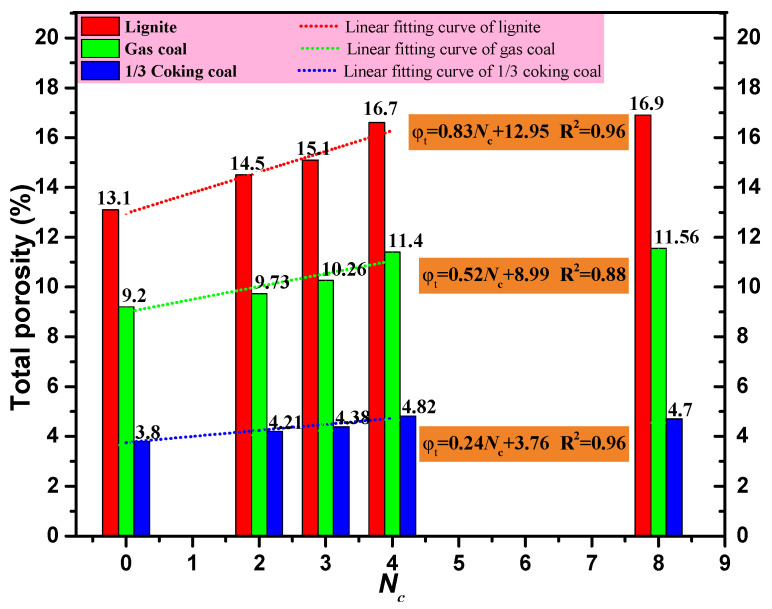
Variation in the total porosity *φ_t_* of the three coal cores.

**Figure 9 ijerph-19-13155-f009:**
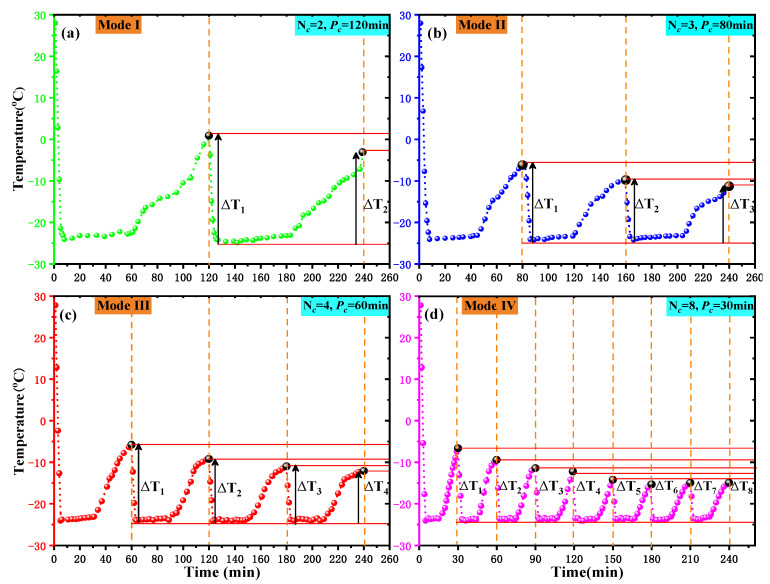
Variation in the effective porosity *φ_e_* of the three coal cores. (**a**) Model I. (**b**) Model II. (**c**) Model III. (**d**) Model IV.

**Figure 10 ijerph-19-13155-f010:**
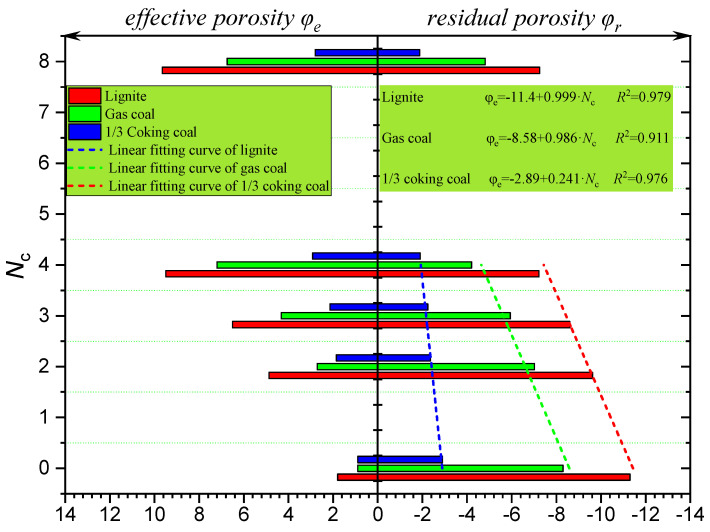
Variation in the effective porosity *φ_e_* and residual porosity *φ_r_* of the three coal cores.

**Figure 11 ijerph-19-13155-f011:**
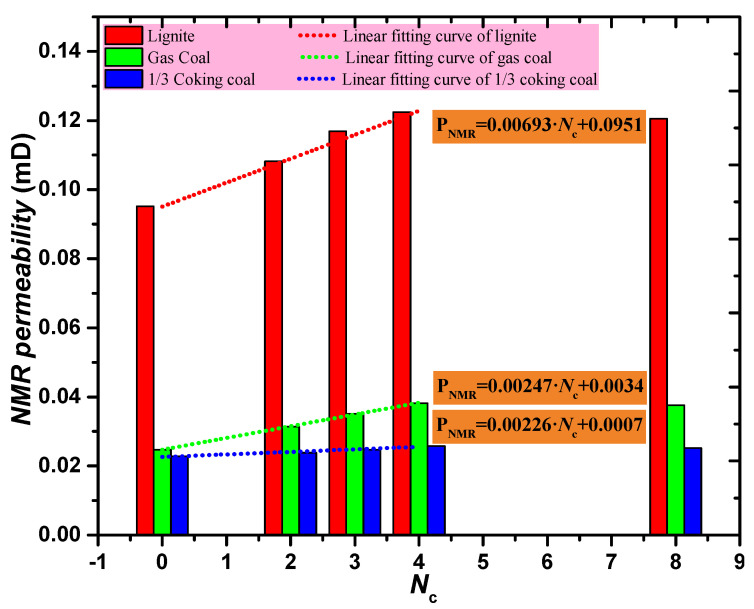
NMR Permeability for the three types of coal cores.

**Figure 12 ijerph-19-13155-f012:**
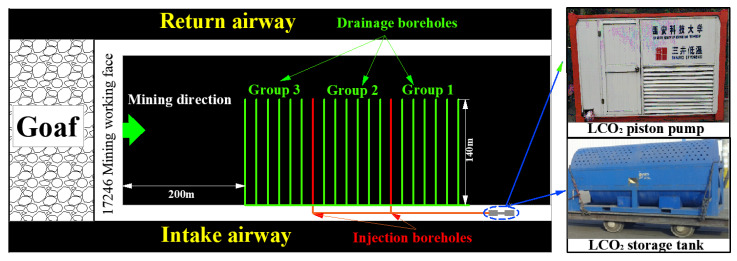
Layout schematic of boreholes in the 17,246 Intake airway.

**Figure 13 ijerph-19-13155-f013:**
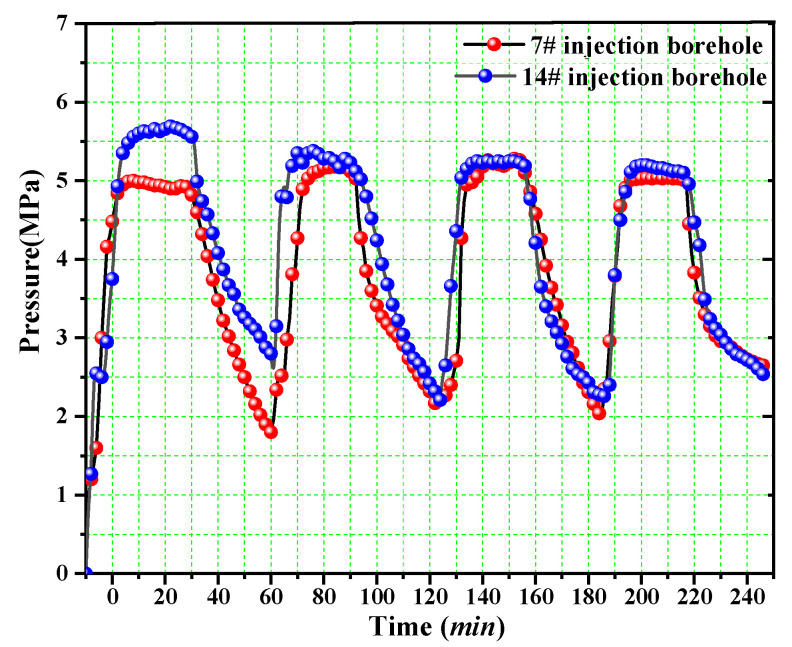
Variation in the pressure of the 7# and 14# injection boreholes.

**Figure 14 ijerph-19-13155-f014:**
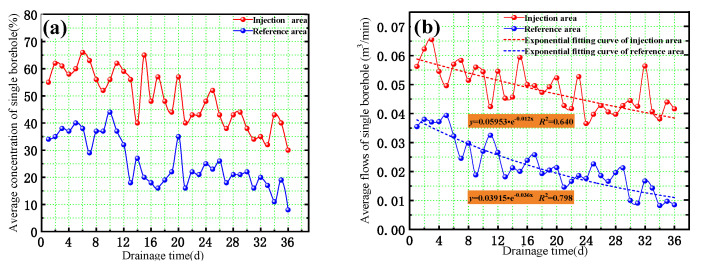
Gas drainage quantity of the drainage boreholes. (**a**) Average concentration of single borehole, (**b**) Average flow of single borehole.

**Table 1 ijerph-19-13155-t001:** Proximate, ultimate analyses, and main features of coal cores used in this study.

Core	Coal Type	Diameter (mm)	Length (mm)	Density (kg/m^3^)	Proximate Analysis (*wt*%)				Water-Saturated Porosity (%)
					*M* _ad_	*A* _ad_	*V* _ad_	*FC* _ad_	
DNH	Lignite	9.9	19.8	1.23	8.80	36.62	24.68	29.90	11.3
ZL	Gas coal	10.0	20.1	1.36	2.06	8.75	32.28	56.91	9.45
ZJ	1/3 Coking coal	9.9	19.9	1.41	1.64	11.35	32.70	54.31	7.38

Note: DNH—Da Nanhu coal mine, ZL—Zhaolou coal mine, ZJ—Zhangji coal mine.

**Table 2 ijerph-19-13155-t002:** Parameters of fitting functions between the total porosity and cycle numbers for three coal cores.

Core	Coal Type	Relationship between *φ_t_* and *N*_c_ Fitting Functions	*a*	*b*	*R* ^2^
DNH	Lignite	*φ_t_* = *a*·*N*_c_ + *b*	0.83	12.95	0.96
ZL	Gas coal		0.52	0.88	0.88
ZJ	1/3 Coking coal		0.24	0.96	0.96

## Data Availability

Research data are not shared. The data that support the findings of this study are available from the corresponding author upon reasonable request.

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
