# Peer review of "Enhancing Methane Recovery with Cryogenic Liquid CO2 Cyclic Injection: Determination of Cyclic Injection Parameters"

_ijerph, 2022, doi:10.3390/ijerph192013155_

Round 1
Reviewer 1 Report
Comments:
Carbon dioxide (CO2) is both primary greenhouse gas and a readily available energy source. This work proposes a new underground coal permeability enhancement technique utilizing cryogenic liquid CO2 (L-CO2) cyclic injection. The key parameters that determine the feasibility of this technique are cycle period and cycle number within a fixed working period. The optimal value of these two parameters mainly depends on the pore structure evolution law of coal cores before and after cryogenic L-CO2 cyclic freeze-thaw. Accordingly, nuclear magnetic resonance (NMR) was employed to study the evolution characteristics of the fracture networks and pore structures in coal cores subjected to different freeze-thaw cyclic modes.
The amplitude and width of all peaks of the T2 spectra of the three coal cores increased with an increase in the number of injection cycles. Additionally, as the number of freeze-thaw cycles increased, the total porosity and effective porosity of the coal cores increased linearly before stabilizing, while the residual porosity first steadily diminished and afterward became consistent. Furthermore, the variation in the total porosity and residual porosity of the coal cores continuously diminished with an increase in the level of metamorphism.
From my side, this work is interesting and timely, but it should be revised carefully further before being reconsidered.
1) Introduction, Line 32., “Gas disasters are one of the most common factors affecting coal mine safety in China:..” The “in China” can be deleted, because the Gas disasters are one of the most common factors affecting coal mine safety for All human beings.
2) Figure 6 & Figure 7 & Figure 14. The pictures are too blurry. Please replot them.
3) Conclusion, Line 433. “The main conclusions are as follows:” It can be deleted. Make no sense.
4) References. The References should be revised carefully according to the author guide of the journal. Please pay attention to the superscript and subscript. (CO2. ref. 3, ref. 19)
5) Introduction. The authors are suggested to further improve the Introduction to highlight the novelty and contribution of this research. From my side, the L-CO2 exploitation of shale gas or Coalbed methane can achieve more in one stroke. Such as improving the efficiency of CBM recovery, and CO2 storage (Permanent geological isolation) to address the global warming issue. Besides, more related and good ref. should be included.
6) Page 2. Line 64-75. “Previous studies have mainly focused on the effects of the huge phase energy of L-CO2 on the coal body. Chen et al. 21 derived the sphere of influence of this technology by combining numerical simulation and underground industrial experiments. Cao et al 22 conducted the industrial test of the L-CO2 phase change fracturing technology in Luan coalfield and found that the permeability and methane drainage effectiveness in the L- CO2 fracturing boreholes were significantly improved. Lu et al. 23 determined that using L-CO2 as a fracturing fluid reduces the fracture initiation pressure (FIP) by more than 30%. Jiang et al. 24 developed an L-CO2 phase-change gas jet pressure model, which was used to explore the mechanism of L-CO2 phase jet coal breaking and fracture expansion. Yin et al. 15, 25 concentrated on the effects of geological and engineering factors on induced fracture propagation by performing hydraulic and L-CO2 fracturing experiments on coal specimens gathered from the South Sichuan Basin in China” Here, please make it clear why you did this work. What is the novelty? Why it is important?
7) How to detect gas disasters in the complex working environment in coal mines? The authors can read this paper [1] Coal Mine Personnel Safety Monitoring Technology Based on Uncooled Infrared Focal Plane Technology[J]. Processes, 2022, 10(6): 1142.
8) Page 2. Line 80. “Upon first injecting gaseous CO2 into the borehole, the drilling pressure increases to 2.0 MPa…” From my side, did you consider the source of CO2, which is also the major greenhouse gas? Where can you get so much gaseous CO2? Kind suggestion, the CO2 can be captured from ambient air (DAC) ([2] Sorbents for the direct capture of CO2 from ambient air[J]. Angewandte Chemie International Edition, 2020, 59(18): 6984-7006.) or it can be captured from the flue gases like power plants, steel works ([3] Structure and surface insight into a temperature-sensitive CaO-based CO2 sorbent[J]. Chemical Engineering Journal, 2022, 435: 134960. [4] CO2 capture at medium to high temperature using solid oxide-based sorbents: Fundamental aspects, mechanistic insights, and recent advances[J]. Chemical Reviews, 2021, 121(20): 12681-12745.).
9) There are too many symbols in the formula of the text, please give a special NOMENCLATURE in the manuscript.
Reviewer 2 Report
The article explores the use of cyclic injection of cryogenic liquid CO2 in coal cores with the aim of improving permeability and methane recovery. Three types of coal cores and different injection cycles were used. The results indicate that the applied method improves the average gas drainage concentration by 1.93 times and the pure flow of single gas drainage borehole by 2.21 times. The authors conclude that the technique used allows transforming coal seams from hard-to-drain to drainable.
General comments:
The article focuses on a relevant topic in view of the growing technical and economic difficulties of methane extraction and the environmental and human risks it involves. The article has very good quality. The Abstract adequately summarizes the work carried out, the results obtained and the conclusions. The introduction provides the necessary and up-to-date framework with very recent literature references. The methodologies used are scientifically sound and are explained with the necessary detail in the experimental section, the usefulness of the figures to be highlighted. The results are clearly presented, except for a few figures (see specific comments below) but are discussed without reference to the literature, although supporting the final conclusions.
Specific comments:
1) In the Abstract, consider presenting the values ​​of the optimal cycle of page 13 lines 364-367, as this is an important result of the work.
2) In the Abstract indicate that 3 types of coal cores were used and which types.
3) In discussing the results, consider presenting and comparing values ​​from the literature for other methods of improving the permeability of coal cores and gas extraction.
4) Improve the legends of Figures 6, 7 and 14 as they are not legible.
